# The Cognition Problem of Surroundings for the Agent Based on Direction Measurement

1st Yingjing Shi
*School of Automation Engineering*
*University of Electronic Science and Technology of China*
Chengdu, P. R. China
shiyingjing@uestc.edu.cn

2nd Rui Li
*School of Automation Engineering*
*University of Electronic Science and Technology of China*
City, Country
lirui@uestc.edu.cn

*Abstract*—**This paper investigates the cognition problem of surroundings for the agent only with the knowledge of angle measurement. In this paper, in virtue of the local motion information of the agent, the problem determining the agent's position and attitude in surroundings with only angle measurement is attributed to solve linear algebraic equations. Based on the obtained position and attitude information, the localization problem of the object in surroundings is handled by solving overdetermined linear algebraic equations. Finally, the proposed methods are verified by the experiments.**

*Index Terms*—**angle measurement, environmental cognition, rotation transformation, overdetermined equation**

## I. INTRODUCTION

For the past decade, cooperative control of multi-agent systems has been an active research area. As one of the fundamental cooperative control problems, the localization problem has been extensively studied. Potential applications of localization algorithms can be found in security and surveillance, and environmental monitoring and sampling. A crucial task in these applications is to find the (possibly time-varying) locations of all the agents based on sensor measurement data available to the agents.

Based on the type of measurement data available, localization algorithms can be classified into three categories: position-based schemes [1]–[5], distance-based schemes [6]–[10] and direction-based scheme. Due to the complex sensing hardware or global positioning system, full information may not be used in some environment. In the distance-based schemes, the relative distances between certain neighboring agents are available for localization purposes. In comparison, the direction-based localization schemes use angle measurements instead of relative distances, which are often cheaper and more accessible than position measurements. Thus, the localization problem has been widely addressed.

In [11], the problem of bearing-only localization of a single target is formulated as a constrained optimization problem using synchronous measurements from multiple sensors, which is solved by an iterative algorithm. In [12], self-localization and formation control tasks are considered when the agents have limited sensing capabilities. [13] considers the problem of localization and circumnavigation of a slowly drifting target with unknown speed using the agents known position and the bearing angle of the target. In [14], the authors further investigate the scenario where the agent moves with constant speed using bearing measurements. [15] investigates the problem of bearing measurement based distributed localization for sensor networks where a common sense of north is not be shared. [16] develops a distributed source localization scheme based on bearing angle measurements for a group of unicycle-type agents without the need of GPS and compass.

In practical application, the global coordinate frame shared among all the agents are required for coordination control of the multi-agent systems. However, since data received from each agent is represented in its own local frame, there is no global coordinate frame shared among all the agents. Therefore, how to construct the transformation relationship between local coordinate frame and its global frame is needed. Most literatures assume that the coordinate transformation relationship mentioned above is known by default or the relative position information can be directly detected, and then focus on developing control algorithms, thus ignoring the systems distributed characteristic. However, in practice, how to obtain the transformation relationship is challenging, especially when the system is based on incomplete information where the sensors can only measure bearing information. In [17], authors discussed preliminarily the calibration problem of the transformation relationship between local coordinate frames. However, localizing the object in the surroundings is not considered.

In this paper, we first investigate how to determine the agent's position and attitude in surroundings with only angle measurement, which is transformed to solve linear algebraic equations. Based on the obtained position and attitude information, the localization problem of the object in surroundings is attributed to solve overdetermined linear algebraic equations. Finally, we develop experiments to verify the availability of the proposed methods.

The rest of this paper is organized as follows. In Section II, objectives of this paper are introduced formally. In Section III, main results of this paper are presented. We first solve how to determine the relationship between the agent and the

This work is supported by the National Natural Science Foundation of China (Grant 61973055); and Natural Science Foundation of Sichuan Province of China (Grant 2023NSFSC0511).

surroundings. Then the localization problem of the object in surroundings is solved. Experiments results are shown in Section IV. Conclusions are included in Section V.

## II. PRELIMINARIES AND PROBLEM STATEMENT

### A. Preliminaries

The theoretical analysis of the paper relies on technique of the coordinate transformation and the linear equation.

Denote $\Sigma_s$ is the surrounding frame and $\Sigma_a$ is the local frame of agent $\nu$. From the knowledge of coordinate transformation, we know there exist rotation matrix $\boldsymbol{R}$ ($\boldsymbol{R}$ is orthogonal) and translation vector $\boldsymbol{t}$, s.t.,

$$\boldsymbol{y}_p = \boldsymbol{R}\boldsymbol{x}_p + \boldsymbol{t}.$$

Here, $\boldsymbol{x}_p$ and $\boldsymbol{y}_p$ are the coordinate of point $P$, respectively, in $\Sigma_s$ and $\Sigma_a$.

Suppose there are $n$ groups of points $\{\mu_i, \nu_i\}$, where $\mu_i$ and $\nu_i$ are, respectively, the points in $\Sigma_s$ and $\Sigma_a$. Here, $\mu_i$ and $\mu_j$ ($i \neq j$) can be the same or not, and $\nu_i$ and $\nu_j$ ($i \neq j$) are as well, but $\mu_i$ and $\nu_i$ ($i = 1, 2, \cdots, n$) are different. Collect the data $\boldsymbol{x}_{\mu_i}$, $\boldsymbol{y}_{\nu_i}$ and $\boldsymbol{l}_i$ ($i = 1, 2, \cdots, n$). Here, $\boldsymbol{x}_{\mu_i}$ is the coordinate of point $\mu_i$ in $\Sigma_s$, $\boldsymbol{y}_{\nu_i}$ is the coordinate of point $\nu_i$ in $\Sigma_a$, $\boldsymbol{l}_i \in \mathbb{R}^3$ denotes the unit vector (in $\Sigma_a$) pointing from the point $\nu_i$ to the point $\mu_i$, that is,

$$\boldsymbol{l}_i = \frac{\boldsymbol{R}\boldsymbol{x}_{\mu_i} + \boldsymbol{t} - \boldsymbol{y}_{\nu_i}}{\|\boldsymbol{R}\boldsymbol{x}_{\mu_i} + \boldsymbol{t} - \boldsymbol{y}_{\nu_i}\|}.$$

Then we can build the nonhomogeneous linear equation as below,

$$\boldsymbol{Q}\boldsymbol{x}_{\mu_i} + \boldsymbol{s} - \lambda_i \boldsymbol{l}_i = \boldsymbol{y}_{\nu_i}, \quad i = 1, 2, \cdots, n \quad (1)$$

where $\boldsymbol{Q} \in \mathbb{R}^{3 \times 3}$, $\boldsymbol{s} \in \mathbb{R}^3$ and $\lambda_i \in \mathbb{R}$ are unknown variables.

**Proposition 1.**[17] $\boldsymbol{R}$ and $\boldsymbol{t}$ are always a solution of equation (1). Another way to think about it is if equation (1) has unique solution $\boldsymbol{Q}_0$ and $\boldsymbol{s}_0$, then $\boldsymbol{Q}_0 = \boldsymbol{R}$ and $\boldsymbol{s}_0 = \boldsymbol{t}$.

**Proposition 2.**[17] In order to solve $\boldsymbol{R}$ and $\boldsymbol{t}$, we only need collect data to guarantee that homogeneous linear equation

$$\boldsymbol{Q}\boldsymbol{x}_{\mu_i} + \boldsymbol{s} - \lambda_i \boldsymbol{l}_i = \boldsymbol{0}. \quad i = 1, 2, \cdots, n \quad (2)$$

has only zero solution.

**Remark 1.** Considering $\boldsymbol{R}^{\mathrm{T}} \boldsymbol{R} = \boldsymbol{I}$ is not a linear relationship, we choose to ignore the orthogonality of $\boldsymbol{R}$ and set up general linear equations, since the linear equations are easy to build and to solve.

**Proposition 3.**[17] Ignore the orthogonality of $\boldsymbol{R}$, if equation (2) has only zero solution, then $\boldsymbol{x}_{\mu_1}$, $\boldsymbol{x}_{\mu_2}$, $\cdots$, $\boldsymbol{x}_{\mu_n}$ must be non-coplanar.

### B. Problem Statement

The objective of this paper includes two parts: (i) Determine the position and attitude relationship between the agent and the surroundings; (ii) Localize the object in the surroundings. Therefore, we propose the following two problems.

**Problem 1.** In 3D space, the goal is to figure out the coordinate transformation relationship between the surrounding frame $\Sigma_s$ and the local frame $\Sigma_a$, that is, to figure out $\boldsymbol{R}$ and $\boldsymbol{t}$.

**Problem 2.** In 3D space, the goal is to figure out the relative relation among the points in surrounding frame, that is, figure out the coordinate of the point in $\Sigma_s$.

## III. PROPOSED SOLUTION

In order to satisfy Proposition 3, the following assumptions are proposed.

**Assumption 1.** $\boldsymbol{x}_{\mu_1}$, $\boldsymbol{x}_{\mu_2}$, $\boldsymbol{x}_{\mu_3}$, $\boldsymbol{x}_{\mu_4}$ are known in $\Sigma_s$, and they are non-coplanar.

**Remark 2.** From the logic of the paper, Assumption 1 is unnecessary. First, the conclusion that "not coplanar is the necessary condition for the existence of only zero solution for (2)" is proposed when the orthogonality of $\boldsymbol{R}$ is ignored. Second, even the above conclusion holds with the orthogonality of $\boldsymbol{R}$ being considered, we didn't mention that (2) has only zero solution is the necessary condition for solving $\boldsymbol{R}$, $\boldsymbol{t}$.

**Assumption 2.** For $t = t_1$ and $t = t_2$ ($t_1 < t_2$), $\nu$ knows the rotation matrix and the translation vector from $t_1$ to $t_2$ (in $\Sigma_a$).

### A. Determine the Relationship between the Agent and the Surroundings

Represent the local frame of agent $\nu$ as $\Sigma_a^T$, when $t = T$. This subsection will determine the relationship between $\Sigma_s$ and $\Sigma_a^T$.

**Theorem 1.** *Suppose that Assumption 1 and Assumption 2 hold, and $\nu$ can measure the directions of four points in Assumption 1. Then Problem 1 is almost always solvable.*

*Proof.* Collect data at $t = 0$ and $t = T$, where $T > 0$. Suppose that the directions of points measured at $t = 0$ by $\nu$ in Assumption 1 are, respectively, $\boldsymbol{l}_1^0, \boldsymbol{l}_2^0, \boldsymbol{l}_3^0, \boldsymbol{l}_4^0$ (in $\Sigma_a^T$), and the directions of that at $t = T$ are, respectively, $\boldsymbol{l}_1^T, \boldsymbol{l}_2^T, \boldsymbol{l}_3^T, \boldsymbol{l}_4^T$ (in $\Sigma_a^T$ too). Denote

$$\begin{bmatrix} \boldsymbol{x}_{\mu_1} & \boldsymbol{x}_{\mu_2} & \boldsymbol{x}_{\mu_3} & \boldsymbol{x}_{\mu_4} \\ 1 & 1 & 1 & 1 \end{bmatrix}$$

as $\boldsymbol{X}$, then from equation (2) we have

$$\begin{aligned} \begin{bmatrix} \boldsymbol{Q} & \boldsymbol{s} \end{bmatrix} \boldsymbol{X} &= \begin{bmatrix} \lambda_1^0 \boldsymbol{l}_1^0 & \lambda_2^0 \boldsymbol{l}_2^0 & \lambda_3^0 \boldsymbol{l}_3^0 & \lambda_4^0 \boldsymbol{l}_4^0 \end{bmatrix} \\ &= \begin{bmatrix} \lambda_1^T \boldsymbol{l}_1^T & \lambda_2^T \boldsymbol{l}_2^T & \lambda_3^T \boldsymbol{l}_3^T & \lambda_4^T \boldsymbol{l}_4^T \end{bmatrix}. \end{aligned}$$

Obviously, for the agent who is freely moving in 3D space, $\boldsymbol{l}_i^0 = \boldsymbol{l}_i^T$ ($i = 1, \cdots, 4$) is an Zero-Probability event, which

means that $\lambda_i^0 = \lambda_i^T = 0$ ($i = 1, \cdots, 4$) is almost always true. Consequently,

$$\begin{bmatrix} Q & s \end{bmatrix} X = 0$$

is almost always true. As can be seen from the premise of the theorem, $X$ is nonsingular, thus $Q = 0$ and $s = 0$ also are almost always true. From Proposition 2, Problem 1 almost always solvable. $\qquad\square$

**Remark 3.** If premise is added to Theorem 1, which is "the agent does not move toward the observed points, that is, $l_i^0 \neq l_i^T$ ($i = 1, \cdots, 4$)", then Problem 1 must be solvable.

In the following, an algorithm is given to solve Problem 1.

**Algorithm 1.** Represent the rotation matrix and the translation vector in Assumption 2 as $\Gamma$ and $\tau$ respectively, that is, $y_T = \Gamma y_0 + \tau$. Then, $R$ and $t$ in Problem 1 can be solved by following steps:

1) Calculate

$$\begin{bmatrix} l_1^0 & l_2^0 & l_3^0 & l_4^0 \end{bmatrix} = \Gamma \begin{bmatrix} l_1(0) & l_2(0) & l_3(0) & l_4(0) \end{bmatrix},$$

where $l_i(0)$ ($i = 1, \cdots, 4$) is the direction recorded at $t = 0$.

2) Calculate $y_s^0 = \Gamma y_s + \tau$, where $y_s$ is installation position of camera on the agent, that is, the coordinate of the camera position in $\Sigma_a^T$.

3) Solve

$$\begin{cases} Q x_{\mu_i} + s - \lambda_i^0 l_i^0 = y_s^0, \\ Q x_{\mu_i} + s - \lambda_i^T l_i^T = y_s. \end{cases} \quad i = 1, \cdots, 4$$

Then, the solution of $Q$ and $s$ are, respectively, the rotation matrix and the translation vector between $\Sigma_s$ and $\Sigma_a^T$.

**Remark 4.** Obviously, the rotation matrix and the translation vector between $\Sigma_s$ and $\Sigma_a^0$ can be obtained by $R$, $\Gamma$, $t$ and $\tau$ (are, respectively, $\Gamma^{-1} R$ and $\Gamma^{-1}(t - \tau)$).

*B. The Localization of the Object in Surroundings*

When the rotation and the translation form $\Sigma_a$ to $\Sigma_s$ are known, that is, $R$ and $t$ have been figured out, then we have Theorem 2 and Theorem 3 as follows.

**Theorem 2.** *Suppose that Assumption 2 holds, $\mu$ is a point in $\Sigma_s$, and $\nu$ is able to measure the direction of $\mu$. Thus Problem 2 is almost always solvable.*

*Proof.* Consider that $\nu$ collects data at $t = 0$ and $t = T$. Denote the rotation matrix and the translation vector from $\Sigma_a^0$ to $\Sigma_a^T$ as $\Gamma$ and $\tau$. The direction of $\mu$ measured by $\nu$ are, respectively, denoted as $l^0$ and $l^T$ (in $\Sigma_a^T$). Let $x$ denote the coordinate of $\mu$ (in $\Sigma_s$). Then we can build the nonhomogeneous linear equation as below,

$$\begin{cases} Rx + t - y_s^0 = \lambda^0 l^0, \\ Rx + t - y_s = \lambda^T l^T. \end{cases} \quad (3)$$

Here, $y_s^0 = \Gamma y_s + \tau$ and $l^0 = \Gamma l(0)$. Nonhomogeneous linear equations (3) have unique solution if and only if homogeneous linear equations

$$\begin{cases} Rx = \lambda^0 l^0, \\ Rx = \lambda^T l^T \end{cases}$$

have only zero solution. Obviously, when $l^0$ and $l^T$ are non-collinear, above equations have only zero solution. Since $l^0$ and $l^T$ are non-collinear almost always true (as long as $\mu$ is not in the direction of $\nu$ motion), the conclusion of the theorem follows readily. $\qquad\square$

In Theorem 2 the target is localized in virtue of the movement of the agent. Due to the relativity of the motion, the following result can be obtained.

**Theorem 3.** *Suppose that the position change of $\mu$ in $\Sigma_s$ from $t = 0$ to $t = T$ is known, and $\nu$ can measure the direction of $\mu$. Then Problem 2 is almost always solvable for stationary $\nu$.*

*Proof.* Represent the coordinate of $\mu$ at $t = 0$ and $t = T$ as $x^0$ and $x^T$ respectively, and let $x^T - x^0 = \sigma$ (in $\Sigma_s$). The directions of $\mu$ measured by $\nu$ at $t = 0$ and $t = T$ are, respectively, denoted as $l^0$ and $l^T$ (in $\Sigma_a$). Then we can build the nonhomogeneous linear equation as below,

$$\begin{cases} Rx^0 + t - y_s = \lambda^0 l^0, \\ Rx^T + t - y_s = \lambda^T l^T, \end{cases} \quad (4)$$

where $x^T = x^0 + \sigma$. This nonhomogeneous linear equations have unique solution if and only if homogeneous linear equations

$$\begin{cases} Rx^0 = \lambda^0 l^0, \\ Rx^0 = \lambda^T l^T \end{cases}$$

have only zero solution. When $l^0$ and $l^T$ are non-collinear, above equations have only zero solution. Since $l^0$ and $l^T$ are non-collinear almost always true (as long as $\mu$ doesnot move in the direction of $l^0$), the conclusion of the theorem follows readily. $\qquad\square$

Another way to think about Theorem 3 is the following corollary.

**Corollary 1.** *Suppose that the relative position relationship between $\mu_1$ and $\mu_2$ in $\Sigma_s$ is known, and $\nu$ can measure the directions of $\mu_1$ and $\mu_2$. Then Problem 2 is almost always solvable for stationary $\nu$.*

## IV. EXPERIMENTS

In this section, three experiments are conducted to illustrate the feasibility of the proposed method. A wheeled robot platform developed in our lab which carries a 3D lidar is used as agent $\nu$. It is noteworthy that although the whole map containing both distance and bearing information of the surroundings of $\nu$ can be obtained by the 3D lidar, only the relative bearing information is utilized to mimic the

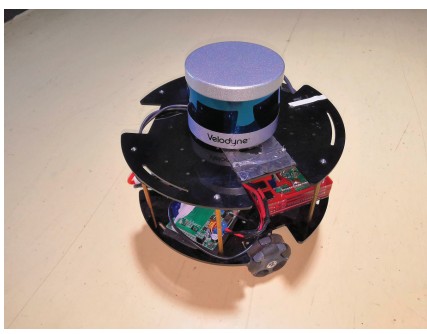

(a) Wheeled robot

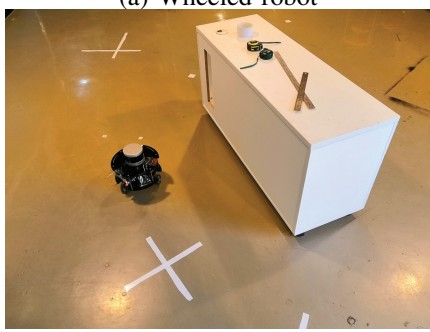

(b) Experiments site

Fig. 1.  The wheeled robot platform and the experiments site.

implementation of the proposed theorems. The wheeled robot platform and the experiments site are shown in Fig. 1.

To verify Theorem 1, Theorem 2 and Theorem 3, Experiment 1, Experiment 2 and Experiment 3 are respectively conducted.

**Experiment 1.**

*Assignments:* In this experiment, the surrounding frame $\Sigma_s$ and the local frame $\Sigma_a$ are defined, respectively, by a cuboidal cabinet (one vertex and its three edges are as origin and three axes of $\Sigma_s$) and the robot's body frame. Four vertices of the cabinet, which are non-coplanar, are used as $\mu_i$ ($i = 1, \cdots, 4$) in surroundings. Therefor, for the data in Theorem 1, what we know ahead are $x_{\mu_i}$, $y_s$, $\Gamma$ and $\tau$, and what we collect are $l_i(0)$ and $l_i^T$. We list these data as TABLE I, and show the map mapping by the 3D lidar in Fig. 2.

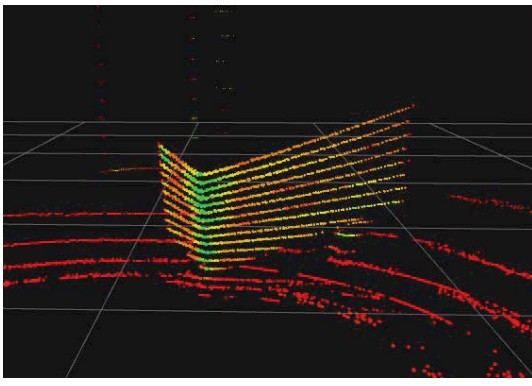

Fig. 2.  The map mapping by the 3D lidar.

TABLE I
THE INPUT DATA.

| Type | Variable | Value |
|---|---|---|
| Know ahead | $\boldsymbol{x}_{\mu_1}$ | $\begin{bmatrix} 36 & 0 & 0 \end{bmatrix}^{\mathrm{T}}$ |
| | $\boldsymbol{x}_{\mu_2}$ | $\begin{bmatrix} 0 & 97 & 0 \end{bmatrix}^{\mathrm{T}}$ |
| | $\boldsymbol{x}_{\mu_3}$ | $\begin{bmatrix} 0 & 0 & 45 \end{bmatrix}^{\mathrm{T}}$ |
| | $\boldsymbol{x}_{\mu_4}$ | $\begin{bmatrix} 0 & 0 & 0 \end{bmatrix}^{\mathrm{T}}$ |
| | $\boldsymbol{y}_s$ | $\begin{bmatrix} 0 & 0 & 17 \end{bmatrix}^{\mathrm{T}}$ |
| | $\boldsymbol{\Gamma}$ | $\begin{bmatrix} 0.839 & -0.545 & 0 \\ 0.545 & 0.839 & 0 \\ 0 & 0 & 1 \end{bmatrix}$ |
| | $\boldsymbol{\tau}$ | $\begin{bmatrix} 21 & -61 & 0 \end{bmatrix}^{\mathrm{T}}$ |
| Collect | $\boldsymbol{l}_1(0)$ | $\begin{bmatrix} 0.494 & 0.849 & 0.189 \end{bmatrix}^{\mathrm{T}}$ |
| | $\boldsymbol{l}_2(0)$ | $\begin{bmatrix} 0.857 & 0.474 & 0.204 \end{bmatrix}^{\mathrm{T}}$ |
| | $\boldsymbol{l}_3(0)$ | $\begin{bmatrix} 0.592 & 0.800 & -0.101 \end{bmatrix}^{\mathrm{T}}$ |
| | $\boldsymbol{l}_4(0)$ | $\begin{bmatrix} 0.569 & 0.803 & 0.181 \end{bmatrix}^{\mathrm{T}}$ |
| | $\boldsymbol{l}_1^T$ | $\begin{bmatrix} 0.115 & 0.959 & 0.260 \end{bmatrix}^{\mathrm{T}}$ |
| | $\boldsymbol{l}_2^T$ | $\begin{bmatrix} 0.683 & 0.683 & 0.258 \end{bmatrix}^{\mathrm{T}}$ |
| | $\boldsymbol{l}_3^T$ | $\begin{bmatrix} 0.332 & 0.930 & -0.157 \end{bmatrix}^{\mathrm{T}}$ |
| | $\boldsymbol{l}_4^T$ | $\begin{bmatrix} 0.315 & 0.911 & 0.265 \end{bmatrix}^{\mathrm{T}}$ |

*Result:* Taking the steps in Algorithm 1, we can calculate the rotation matrix and the translation vector between $\Sigma_s$ and $\Sigma_a^T$ by input data in TABLE I. Measured values ($\boldsymbol{R}_m$ and $\boldsymbol{t}_m$) and truth values ($\boldsymbol{R}_t$ and $\boldsymbol{t}_t$) are given in TABLE II. The initially solved measured value $\boldsymbol{R}_m$ is not strictly orthogonal, so we use *Schmidt's* method to make it orthogonal. And

$$\boldsymbol{R}_m - \boldsymbol{R}_t = \begin{bmatrix} 0.046 & 0.017 & 0.032 \\ 0.038 & -0.039 & -0.066 \\ -0.064 & -0.008 & -0.018 \end{bmatrix},$$

$$\boldsymbol{t}_m - \boldsymbol{t}_t = \begin{bmatrix} 0.1 \\ 0.6 \\ -0.5 \end{bmatrix}.$$

It is shown that the measured values are close to the truth values.

TABLE II
THE OUTPUT DATA.

| Type | Variable | Value |
|---|---|---|
| Measured values | $\boldsymbol{R}_m$ | $\begin{bmatrix} -0.373 & 0.923 & 0.094 \\ 0.905 & 0.340 & 0.257 \\ 0.205 & 0.181 & -0.962 \end{bmatrix}$ |
| | $\boldsymbol{t}_m$ | $\begin{bmatrix} 27.1 & 79.6 & 41.5 \end{bmatrix}^{\mathrm{T}}$ |
| Truth values | $\boldsymbol{R}_t$ | $\begin{bmatrix} -0.419 & 0.906 & 0.062 \\ 0.867 & 0.379 & 0.323 \\ 0.269 & 0.189 & -0.944 \end{bmatrix}$ |
| | $\boldsymbol{t}_t$ | $\begin{bmatrix} 27.0 & 79.0 & 42.0 \end{bmatrix}^{\mathrm{T}}$ |

**Experiment 2.**

*Assignments:* This experiment is in line with the previous one. The output data in Experiment 1 are served as known conditions in this experiment and another vertex of the cabinet is served as observed point ($\mu$ in Theorem 2). Therefore, for the data in Theorem 2, what we know ahead are $\boldsymbol{R}$, $\boldsymbol{t}$ (that is $\boldsymbol{R}_m$ and $\boldsymbol{t}_m$ in TABLE II), $\boldsymbol{y}_s$, $\boldsymbol{\Gamma}$ and $\boldsymbol{\tau}$ (see

TABLE I), and what we collect are $l(0)$ and $l^T$. Here, $l(0)$ and $l^T$ are, respectively, $\begin{bmatrix} 0.863 & 0.505 & 0.002 \end{bmatrix}^T$ and $\begin{bmatrix} 0.678 & 0.735 & 0.017 \end{bmatrix}^T$.

*Result:* Substitute above input data to equations (3), then we can calculate the coordinate of the point $\mu$. Here measured value $x_{\mu,m}$ and truth value $x_{\mu,t}$ are, respectively, $\begin{bmatrix} 2.1 & 89.9 & 41.1 \end{bmatrix}^T$ and $\begin{bmatrix} 0 & 97 & 45 \end{bmatrix}^T$. And

$$x_{\mu,m} - x_{\mu,t} = \begin{bmatrix} 2.1 & -7.1 & -3.9 \end{bmatrix}^T.$$

The measured value is not so close to the truth value, since for our 3D lidar, it is hard to measure the positions of the expected points exactly.

**Experiment 3.**

*Assignments:* This experiment is also in line with Experiment 1. $\mu$ in Theorem 3 is represented by $\mu_1$ in Theorem 1, and it moves 36 units along the $x$-axis of $\Sigma_s$. Therefore, for the data in Theorem 3, what we know ahead are $\sigma$ (that is $\begin{bmatrix} 36 & 0 & 0 \end{bmatrix}^T$), $R$, $t$ (that is $R_m$ and $t_m$ in TABLE II), $y_s$ (see TABLE I), and what we collect are $l^0$ and $l^T$. Here, $l^0$ and $l^T$ are, respectively, $\begin{bmatrix} 0.315 & 0.911 & 0.265 \end{bmatrix}^T$ and $\begin{bmatrix} 0.115 & 0.959 & 0.260 \end{bmatrix}^T$.

*Result:* Substitute above input data to equations (4), then we can calculate the coordinate of point $\mu$. Measured value $x_{\mu,m}$ and truth value $x_{\mu,t}$ are, respectively, $\begin{bmatrix} 33.2 & -1.9 & 1.4 \end{bmatrix}^T$ and $\begin{bmatrix} 36 & 0 & 0 \end{bmatrix}^T$. And

$$x_{\mu,m} - x_{\mu,t} = \begin{bmatrix} -2.8 & -1.9 & 1.4 \end{bmatrix}^T.$$

It is concluded that the measured value is close to the truth value.

## V. CONCLUSIONS

This study is based on the assumption that the agent can only access the angle information of the surroundings and can not obtain the distance information. First, when the agent is with the sensing ability to know their own motion, we propose a method to solve the position and attitude of the agent in surroundings via four non-coplanar points. Second, with the obtained position and attitude of the agent, we propose a method to localize the objects in the surroundings via the motion of the agent. Finally, we extend the developed methods and establish a way to localize two objects with known relative position by agent. The proposed methods are achieved by solving the linear algebraic equations and are easy to iteratively correct, thus it suitable for the situation where rapid attitude orientation is needed.

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
