# OpenReview forum: "The Cognition Problem of Surroundings for the Agent Based on Direction Measurement"
_IEEE.org/ICIST/2024/Conference — IEEE ICIST 2024 Conference Submission_

### Official Review · Reviewer_LkVo · 2024-08-27
**Accept**

**Rating:** 7
**Confidence:** 5

**Review:**

This paper explores the cognitive challenge of an agent's surroundings when it possesses only angle measurement capabilities. By leveraging the agent's local motion information, the authors effectively frame the problem of determining the agent's position and orientation within its environment solely through angle measurements as a series of linear algebraic equations. Subsequently, utilizing the derived position and orientation data, the localization of objects in the environment is addressed by solving overdetermined linear algebraic equations. While the overall approach is well-conceived and the logic is sound, there are a few areas that could benefit from further refinement:
1. The introduction could provide a more thorough discussion of the limitations of existing methods in addressing similar problems, thus emphasizing the novelty and necessity of the current research.
2.The conclusion should more explicitly highlight the key contributions of the paper and offer insights into potential avenues for future research.

---

### Official Review · Reviewer_x7ZF · 2024-08-27
**summarize the innovation and individual originality work**

**Rating:** 7
**Confidence:** 4

**Review:**

In the manuscript titled "The Cognition Problem of Surroundings for the Agent Based on Direction Measurement"presents a method for determining an agent's position and orientation using only angle measurements and local motion data, which is solved through linear algebraic equations. It also addresses object localization in the environment, with methods validated by experiments.This study contains some interesting findings and is valuable for the understanding of how an agent can determine its position, orientation, and object localization using only angle measurements and known motion, with methods that are effective for rapid adjustments and iterative corrections..However, lack of the innovations and individual originality work is the major flaw of the study .it is suggested to summarize the innovation and individual originality work point by point, in other words, add the content of this section.

---

### Official Review · Reviewer_LmSv · 2024-08-30
**The paper is written clearly, exceptionally excellent.**

**Rating:** 8
**Confidence:** 3

**Review:**

This paper excels in terms of quality, clarity, originality, and significance, but I would still like to offer some suggestions.
1. Please introduce the possible future work.
2. Please provide a brief discussion on the feasibility and rationality of the assumptions made in the paper.

---

### Decision · Program_Chairs · 2024-09-08

Accept (Oral)